# Diversity and Functional Roles of the Gut Microbiota in Lepidopteran Insects

**DOI:** 10.3390/microorganisms10061234

**Published:** 2022-06-16

**Authors:** Xiancui Zhang, Fan Zhang, Xingmeng Lu

**Affiliations:** 1Institute of Sericulture and Apiculture, College of Animal Sciences, Zhejiang University, Hangzhou 310029, China; xczhang525j@zju.edu.cn; 2Key Laboratory of Animal Resistance Biology of Shandong Province, College of Life Science, Shandong Normal University, Jinan 250014, China

**Keywords:** lepidopteran insect, gut microbiota, diversity, function

## Abstract

Lepidopteran insects are one of the most widespread and speciose lineages on Earth, with many common pests and beneficial insect species. The evolutionary success of their diversification depends on the essential functions of gut microorganisms. This diverse gut microbiota of lepidopteran insects provides benefits in nutrition and reproductive regulation and plays an important role in the defence against pathogens, enhancing host immune homeostasis. In addition, gut symbionts have shown promising applications in the development of novel tools for biological control, biodegradation of waste, and blocking the transmission of insect-borne diseases. Even though most microbial symbionts are unculturable, the rapidly expanding catalogue of microbial genomes and the application of modern genetic techniques offer a viable alternative for studying these microbes. Here, we discuss the gut structure and microbial diversity of lepidopteran insects, as well as advances in the understanding of symbiotic relationships and interactions between hosts and symbionts. Furthermore, we provide an overview of the function of the gut microbiota, including in host nutrition and metabolism, immune defence, and potential mechanisms of detoxification. Due to the relevance of lepidopteran pests in agricultural production, it can be expected that the research on the interactions between lepidopteran insects and their gut microbiota will be used for biological pest control and protection of beneficial insects in the future.

## 1. Introduction

Lepidoptera is one of the most widespread and diverse insect clades in terms of the number of species and biomass [1,2]. Approximately 180,000 species of lepidopteran insects have been described all over the world, some of which act as forestry and agricultural pests, such as butterflies, moths, and skippers. They cause severe damage to major crops, and hence, an efficient strategy is required for management; however, adults of many species play a vital role in ecosystems as pollinators and as prey in the food chain. Others, such as *Bombyx mori*, can be regarded as economically important insects [3,4].

The microbiota of insects comprises bacteria, fungi, viruses, archaea, and protozoa, of which bacteria are found in the gut of almost all insects and are often the most abundant microbes [5,6,7]. Additionally, these microbial symbionts can be divided into endosymbionts and ectosymbionts based on whether they live within insect tissue cells or colonize the lumen or lining of insect body surface cavity walls [8]. Microbial inhabitants are pervasive in hosts and have ubiquitous impacts on multiple aspects of insect biology [9,10]. The evolutionary and diversification success of insects into large-scale ecological niches depends on the beneficial microbiome, which is known to promote insect fitness, protect hosts against parasites and pathogens, detoxify insecticidal defence chemicals, and stimulate host immune responses, in addition to its biotechnological applications [11,12,13,14]. For example, gut microbial communities enhance digestive efficiency by providing enzymatic functions and facilitating vitamin synthesis, helping their hosts optimize nutrient absorption and energy extraction [15]. Liang et al. reported that *Enterococcus mundtii*, isolated from the *B. mori* gut, efficiently produces lactic acid under extremely alkaline conditions and is an important metabolite for industrial bioplastic polylactic acid production [16]. In contrast, the lepidopteran microbiome is affected by many factors, including the environment, diet, gut physiology, and developmental stage. Recent studies have found that the microbial community changes significantly between early (1st and 2nd) and late (3rd to 5th) instar silkworms, consistent with host developmental changes [5].

With recent developments in Next-generation sequencing (NGS) technology, a growing number of studies have catalogued and characterized microbial communities [17,18,19]. To date, several studies have examined their role via approaches ranging from community functional diversity surveys to the examination of gut bacterial interaction mechanisms with hosts. However, many of the studies are mostly descriptive and have focused on larvae, while only a few have addressed the potential impact on host traits. Here, we highlight the structure, function, and host relationships of the lepidopteran insect gut microbiota. Based on previous studies of the gut microbiota, we provide some promising new insights into the fundamental molecular mechanisms of insect immunity and integrated pest management applications (i.e., utilizing symbionts to control devastating pests) [20].

## 2. Gut Structure of Lepidopterans

The alimentary canal of the lepidopteran larva is divided into three primary regions: The foregut and hindgut, which arise from the embryonic ectoderm and are lined with the chitin exoskeleton, and the midgut, which originates from the ectoderm [21,22,23]. The demarcations of each part are distinguished by the cardiac valve (between the foregut and midgut) and pylorus (between the midgut and hindgut) (Figure 1).

The foregut of many lepidopteran insects comprises the crop and oesophagus, where food fragments are stored [24,25]. For most lepidopteran larvae, including silkworms, the midgut is observably larger than the foregut and hindgut, secreting the vast majority of enzymes and small molecules for food digestion, such as proteases and carbohydrases. Thus, the midgut is the primary site of digestion and absorption. In addition, the midgut lines the peritrophic matrix, which includes three different cell types: goblet cells, columnar cells, and stem cells [26,27,28]. Goblet cells are considered to differentiate from stem cells, and there is evidence that lepidopteran insect goblet cells have a critical role in the gut immune defence [29,30,31]. The peritrophic matrix distinguishes the midgut into two spaces: endo-peritrophic and ecto-peritrophic. Normally, gut microorganisms cannot cross the endo-peritrophic space, which prevents them from coming in direct contact with midgut epithelial cells [32,33,34]. The hindgut of lepidopteran insects consists of three regions: the ileum, the rectum, and the posterior rectum. They mediate the uptake of uric acid, water, and salts derived from Malpighian tubules, which are part of the excretory system of insects [30,35]. All these structures are beneficial to the caterpillar, leading to very high feeding and food digestion rates [36,37].

Gut physicochemical conditions can influence the metabolic activity of gut microbes, including pH, oxygen availability, redox conditions, ion concentrations, and digestive enzymes [30,38,39]. The pH of the digestive tract is energetically regulated and often departs from that of the haemolymph, which is generally approximately 7 [40,41,42]. The gut of lepidopteran larvae generally exhibits extreme alkalinity, with a midgut pH as high as 7–12 [43]. Compared with that of the midgut, the alkalinity of the foregut and hindgut is relatively weak. Therefore, digestive enzymes in the lepidopteran gut are accustomed to alkaline conditions [44,45]. For example, the activity of intestinal phosphatase as a mucosal defence factor requires an alkaline environment, which is achieved by the proton pump activity of V-ATPase [22,46].

## 3. Diversity of the Lepidopteran Gut Microbiota

Lepidopteran insects harbor a large number of microbiota in their midgut, which includes both pathogenic and nonpathogenic bacteria [47]. Insights into the composition of the species-specific gut microbiota are principally obtained from culture-independent techniques and NGS approaches [48,49]. Similar to other invertebrate and vertebrate species, highly active microorganisms live inside the lepidopteran gut, including bacteria, fungi, and archaea. Paniagua Voirol et al. surveyed 30 lepidopteran species and found that gut bacteria of the Enterobacteriaceae, Pseudomonadaceae, and Bacillaceae families were the most widespread [50]. For example, *Enterobacter*, *Pantoea*, *Pseudomonas*, and *Acinetobacter* were present in *Acronicta major* larvae [5]. Compared to that in *A. major*, the bacterial diversity observed in *Diaphania pyloalis* was observably simple, principally composed of *Wolbachia* (40.60%) [5]. In addition, bacteria detected in the gut microbiota of *B. mori* larvae were distinguished based on high-throughput sequencing. The results show that Proteobacteria, Firmicutes, Actinobacteria, and Bacteroidetes are dominant species [5]. In the intestines of other lepidopteran insects, such as *Plutella xylostella*, high-throughput sequencing revealed that 97% of the bacteria were from Enterobacteriales (45.17%), Vibrionales (22.51%), and Lactobacillales (29.49%) [51]. A core gut community consisting of *Enterococci* (42.3%) and *Clostridium* (42.2%) were revealed in *S. littoralis* larvae, *Helicoverpa armigera* larvae, and *S. littoralis* larvae [52].

Notably, the type of diet, host plant, season, population density, and geographic position influence gut bacterial diversity [53]. Feeding can change the gut microbiota community of lepidopteran insects [52,54]. For instance, mulberry leaves are primarily composed of xylan (10–40%) and cellulose (19–25%), which shows the importance of intestinal microbes for food digestion in silkworms [55]. In 5th-instar larvae of *B. mori* fed on mulberry leaves (the traditional rearing method), the gut microbiota is dominated by *Rhodococcus*, *Escherichia*, and *Enterococcus* [3,56]. When the diet was changed to lettuce leaves, *Bacteroides* and *Acinetobacter* were the predominant species [57]. In addition, the species diversity and richness of the gut microbial communities showed a significant relationship with the *Agrilus planipennis Fairmaire* population size [58]. Furthermore, lepidopteran insects are holometabolic, and few studies have reflected the gut microbiota composition throughout development from egg to adult, especially in monophagous species. Francisco et al. showed that the bacterial composition of *Brithys crini* was stage-specific, and *Rosenbergiella* and *Serratia* were highly abundant in the eggs. Twenty-seven genera (*Empedobacter*, 23%; *Enterococcus*, 10%) were statistically more abundant in larvae, while only one genus (*Serratia*, 75%) was significantly more abundant in adults [59]. More surprisingly, recent work has shown that DNA extraction methodology has the largest effect on the outcome of the metagenomic analysis in *B. mori* gut microbiome studies based on high-throughput 16S rRNA gene sequencing and computational analysis [60]. A taxonomic analysis revealed that the most common phylum was Proteobacteria, which, together with Firmicutes and Actinobacteria, was detected in lepidopteran insects. At the genus level, the dominant bacteria were mainly *Enterococcus*, *Enterobacter*, *Clostridium*, *Acinetobacter*, *Pseudomonas*, *Pantoea*, and *Bacillus*. The composition of the dominant gut microbiota of other insects was different. These differences depend on the diet source and behavioral characteristics of the host insects, which show the relationship between gut symbiotic bacteria and the coevolution of the host from another perspective [61,62].

Although a few sequencing-based studies have confirmed the composition of gut bacteria, lepidopteran fungal communities have been largely ignored. However, endosymbiotic fungi are also ubiquitous among lepidopteran insects. Here, we review the reported fungal gut microbiota of lepidopteran insects, including *Lycaeides melissa*, *A. planipennis*, *A. major*, *D. pyloalis,* and *B. mori.* Basidiomycota and Ascomycota predominated the gut fungal communities, as determined by sequencing of the fungal internal transcribed spacer (ITS). Most fungal sequences were assigned to the genera Ascomycota and Basidiomycota. At the genus level, most fungal sequences were assigned to the genera *Cladosporium*, *Hannaella*, *Kabatiella*, *Pyrenochaeta*, *Pyrenochaeta*, *Malassezia*, and *Rhodosporidium* [5,58,63].

## 4. Functional Roles of the Lepidopteran Gut Microbiota

The success of lepidopteran insects in diversity and evolution depends on various beneficial gut symbiotic bacteria, especially for upgrading nutritionally deficient diets [64]. The limited metabolic networks of most insects have been enhanced by symbiotic relationships. The insect gut is colonized by multitudinous communities of resident bacteria, and such microbes are considered to be essential for the fecundity, development, and growth of the hosts [65,66]. They not only play important roles in food digestion and the production of vitamins but also contribute positively by protecting the host against pathogens, detoxifying insecticidal defence chemicals and stimulating the host immune response [67,68] (Table 1).

### 4.1. Host Nutrition and Metabolism

Insects provide stable environments and nutrition for symbionts, and in return, symbionts can offer the host necessary enzymes for food digestion, thereby expanding the host’s diet options and even changing the host’s eating habits [89]. The symbionts of the gut can contribute to the nitrogen cycle and can also produce nutrients that are essential to the development of the host organisms but are lacking in natural food, including amino acids, B vitamins, and sterols [90]. For instance, 118 culturable bacterial strains were isolated from the intestine of *Diatraea saccharalis* larvae. Among them, *Klebsiella*, *Stenotrophomonas*, *Microbacterium*, *Bacillus*, and *Enterococcus* were found to possess cellulolytic activity. All bacterial strains were cultured using soluble carboxymethyl cellulose (CMC) for degradation assays, and *Bacillus* and *Klebsiella* showed the highest degradation activity [91]. In addition, ten gut bacteria were isolated from the lepidopteran insect gut by in vitro culture, including gram-positive and gram-negative bacteria. *Klebsiella* can hydrolyse starch, whereas *Proteus vulgaris*, *Erwinia sp*., and *Serratia liquefaciens* can utilize xylanolytic, pectinolytic, and polysaccharides, respectively [56]. The main components of mulberry leaves are cellulose (19% to 25%) and xylan (10% to 40%), which shows the importance of the intestinal microbes of silkworms for food digestion [92,93].

Gut symbionts (*Bacillus cereus*, *Enterococcus gallinarum*, *E. mundtii*, *Staphylococcus xylosus*) are the pivotal species in soybean pests and are abundant in the caterpillar host. They exhibit a high tolerance for serine-proteinase inhibitors [70]. *Enterobacter asburiae* YT1 and *Bacillus* YP1 from the larvae of *Plodia interpunctella* were capable of degrading polyethylene films [94]. In addition, vitamins are the fundamental micronutrients that are normally found as precursors of various enzymes that are necessary for vital biochemical reactions during insect growth and development [95]. Hassan et al. tested the hypothesis that two actinobacterial gut symbionts provide *Dysdercus fasciatus* with B vitamins [37]. Insects actively harvest vitamins from bacterial symbionts by using specific enzymes that burst open the bacterial cell walls and thereby ensure host metabolic homeostasis [47].

### 4.2. Pathogen and Immune Defences

Under normal living conditions, the gut is the first line of defence because ingestion is the most likely route by which organisms come in contact with pathogens, including bacteria, fungi, viruses, and parasites [96,97]. Insects lack adaptive immune components, such as B cells and T cells, and rely on innate immune responses against infection. To combat infection, insects rely on multiple innate defence mechanisms, including the use of immune responses together with resource competition [98]. Lepidopteran insect guts use a battery of strategies, such as the generation of reactive oxygen species (ROS), to defend against harmful bacteria through cellular immune responses. Moreover, antimicrobial peptides (AMPs) and other immune effector molecules exhibit a broader spectrum of antimicrobial activity (Figure 2). In addition, the peritrophic membrane is a semipermeable barrier that can prevent most pathogens from damaging gut tissue via *per os* infection [99,100,101].

Bacteria are important pathogens of lepidopteran insects [102]. Bacterial peptidoglycans (PGs) and proteases may disrupt the host’s cellular and biochemical processes [103,104]. The recognition of pathogens by lepidopteran insects relies on the interaction between pathogen-associated molecular patterns (PAMPs) and pattern-recognition receptors (PRRs) [105,106]. Lys (lysine) and lipopolysaccharide (LPS), as immune stimulators in insects, are major components of bacterial cell walls. They can trigger strong host immune responses in multitudinous insects [107,108,109]. In the midgut of lepidopteran insects, the immune reaction is primarily mediated by regulating the expression level of key immune components in the dual oxidase (DUOX) system and the immune deficiency (IMD) pathway, thus obtaining immune tolerance to beneficial gut microorganisms. Larvae carrying a Duox deletion are more susceptible to bacterial infection. Similar to the immune response, the local systemic response is regulated via the recognition of gram-negative proteoglycans (PGNs) by peptidoglycan recognition protein LC (PGRP-LC). Injection of pathogenic bacteria induces transient expression of AMP genes, suggesting the existence of a mechanism to downregulate the host immune response (Figure 2) [110,111]. For example, the expression of BmDuox was significantly upregulated in the midgut of *B. mori* fed *Escherichia coli.* Microbial proliferation in the midgut was increased after BmDuox knockout, suggesting that BmDuox has an important role in maintaining gut microbial homeostasis [112]. Peroxiredoxins (Prxs), as antioxidant enzymes in the lepidopteran insect gut, are notably enriched upon *Pseudomonas aeruginosa* and *Bacillus bombyseptieus* infection, and increased ROS levels can be induced by bacterial infection and proliferation [103,113]. In addition, the immune system of *Hyalophora cecropia* and *Galleria mellonella* was found to contain P9A and P9B antibacterial proteins, which are active against several Gram-negative bacteria (i.e., *Escherichia coli* and *P. aeruginosa*) [114]. In some lepidopteran insects, such as *Choristoneura fumiferana*, depletion of the gut microbiota increases the susceptibility of hosts to pathogenic infection [115]. Intriguingly, some lepidopteran gut microbes are universal opportunistic pathogens [50]. A commensal-to-pathogen switch is observed under multifactorial conditions, which depends on the pathogens and immune status of the host. This poses the question of how the immune system in the gut distinguishes between symbiotic microorganisms and pathogenic bacteria [116].

Fungi, such as *Beauveria bassiana*, *Metarhizium anisopliae*, and Microsporidia, are another group of important pathogens of lepidopteran insects [117]. Transcriptomic analyses revealed that infection by the *B. bassiana* strain upregulated the expression of immunity-related genes in *G. mellonella*, including hydrolytic enzymes, β-1,3-glucan recognition proteins, and spätzle genes [118]. A significant increase in the expression pattern of prophenoloxidase cascade (PPO) genes was found in *Chilo suppressalis* after treatment with *B. bassiana*, *M. anisopliae*, *Isaria fumosoroseus* and *Lecanicilium lecanii*, suggesting that host immune responses are critical against fungal infections [119]. Another study predicted serine proteases (SPs) and pattern recognition receptors (PRRs) as upstream components of the Toll pathway in *Manduca sexta* and *Spodoptera exigua* infected with *Metarhizium rileyi* [120]. In addition, Microsporidia, which are pathogens of lepidopteran insects, are a group of obligate intracellular parasites related to fungi. *N. bombycis* mainly infects *B. mori* through oral infection, and cuticle infection occasionally occurs [117]. Virulence studies showed that *per os* infection of silkworm larvae by microsporidia led to stimulation of the JAK/STAT and Toll signalling pathways in the midgut, which possibly induced the upregulation of AMPs to defend against the invading *N. bombycis*. The subtilisin-like serine protease NbSLP1 was activated after infection of *N. bombycis* in the midgut [121]. NbSLP1 is localized at the two poles of the spore and is likely involved in the polar tube extrusion process [122]. Two studies have also shown that feeding with *Enterococcus faecalis* LX10 or *Lactobacillus* could reduce the spore germination rate or increase the survival rate of silkworm larvae challenged by *N. bombycis* [3,123].

Viruses are significant natural pathogens of lepidopteran insects, and horizontal transmission of viruses is common in these species [124]. In addition, viruses infecting beneficial insects such as silkworms or bees can cause significant economic losses [125]. Host responses to viral infections include immunoreactions as well as mechanical barriers that prevent viruses from establishing infection [126]. Agata et al. observed that baculovirus infection leads to decreased expression of immune genes in the *S. exigua* larval gut. The expression of immune genes affects the diversity of gut microorganisms, many of which are responsible for growth and development functions [126]. In addition, several immune-related genes were found to be implicated in the midgut’s response against BmCPV infection of *B.*
*mori* larval, including proteolytic enzymes, hormonal signaling, and heat-shock proteins [127]. In *Drosophila*, RNAi is a powerful method for defending against viruses, and activation of the Toll pathway inhibited Drosophila virus growth [128]. In the midgut of *B. mori*, alkaline trypsin protein and serine protease-2 showed strong antiviral activity, while immunoglobulin proteins, including Hemolin, a lepidopteran plasma protein produced after viral injection, demonstrated antiviral activity in oak silkworm, *M. sexta* and the *Samia cecropia* [129,130]. These studies indicate that lepidopterans circulate key proteins that serve as potent antiviral factors in the midgut.

Recent studies have shown that gut microorganisms can protect insects from propagating pathogens by accommodating host metabolism and repairing gut wall integrity, stimulating the host immune system and serving as essential probiotics for insect growth and development [131,132].

### 4.3. Potential Mechanism of Detoxificationby Lepidopteran Gut Bacteria

Gut symbiotic bacteria can also assist the host in degrading toxic or harmful substances, including insecticides, secondary plant compounds, and microplastics.

It has been reported that bacteria can directly degrade organic insecticides, such as ethoprophos, dimethoate, and chlorpyrifos, and these bacteria are often ingested from sources in the environment and food sources by agricultural pests [133,134,135]. Moreover, the gut microbiota may also enhance detoxification by influencing host fitness and the immune system [136]. For example, some soil *Burkholderia* strains degrading fenitrothion establish symbiosis with *Riptortus pedestris* and enhance host resistance to fenitrothion [137]. Indoxacarb is a highly effective insecticide widely used in the production of fruits and vegetables. *B. cereus* from the *P. xylostella* (Linnaeus) gut microbiota degraded indoxacarb by up to 20% and could use insecticides as an energy substance for growth and metabolism [138]. In addition, monoassociation of *B. mori* with gut bacteria of the genus *Stenotrophomonas* enhanced host resistance to organophosphate insecticides (chlorpyrifos), as confirmed by gut metabolomic analysis [83].

The majority of plants produce a wide variety of secondary metabolites that are toxic to pathogens and herbivores [139]. Recent studies have shown that gut microorganisms can assist the host in degrading toxic secondary metabolites. For instance, the gut bacteria *Acinetobacter* sp. R7-1 of *Lymantria dispar* has already been confirmed to metabolize aspen foliage secretion (phenolic glycosides) [140]. In particular, *Klebsiella* sp. and *Corynebacterium* have been isolated from the polyphagous pest larvae of *Brithys crini*, which participate in the degradation of alkaloids [141]. Gut bacteria protect *Trichoplusia ni* and *Spodoptera eridania* from the host plant toxin hydrogen cyanide (HCN) [142]. Some gut bacteria of *Trichoplusia ni* and *S. eridania* are capable of detoxifying toxic HCN, producing β-cyanoalanine (nontoxic product) and cysteine [143]. In addition, the *E. casseliflavus* strain was isolated from the gut and exhibited the ability to tolerate natural latex under laboratory conditions [141]. Xia et al. revealed an important role of *Enterobacter cloacae*, *E. asburiae*, and *Carnobacterium maltaromaticum* in the breakdown of plant cell walls, detoxification of plant phenolics, and synthesis of amino acids of the *P. xylostella* gut [144]. Members of the genera *Pseudomonas*, *Burkholderia*, and *Cupriavidus* were selected from the moth *Retinia resinella* and exhibited the ability to degrade specific resin acids such as dehydroabietic or isopimaric acid (diterpenes) [145].

In addition, several polyethylene (PE)-degrading bacteria and fungi have been reported, such as *Aspergillus*, *Acremonium*, *Fusarium*, *E. asburiae*, and *Bacillus* [146]. PE is one of the polymer materials that are remarkably resistant to degradation [147]. A fungal strain, *Aspergillus flavus*, was isolated as a potential microplastic particle-degrading microorganism from the gut contents of wax moth *G. mellonella* larvae by producing extracellular enzymes [148]. *E. asburiae* and *Bacillus* strains isolated from the gut of *P. interpunctella* can degrade polyethylene by forming biofilms that reduce the hydrophobicity of PE [94]. Subsequently, Ren et al. isolated the *Enterobacter* sp. strain D1, with the ability to degrade PE films, from the gut of *G. mellonella* (Ren et al., 2019). Recently, *B. mori* has also been applied in nanotoxicology studies to assess the potential effects of TiO_2_ nanoparticles on intestinal microbes [149]. These results suggested that the gut of insects might serve as a potential source for selecting PE-degrading microorganisms. It may also be possible to develop new strategies to reduce the toxic effects of xenobiotics on insects by leveraging their microbial symbionts.

### 4.4. Potential Application of Gut Symbionts in Controlling Lepidopteran Pests

The frequent application of insecticides has led to the ongoing development of high resistance in the past two decades, enhancing the urgent need for environmentally friendly long-term alternative strategies to control them [6,150]. Recent studies have shown that bacterial symbionts constitute promising microbial control agents (MCAs) with potential applications in controlling major lepidopteran agricultural pests, such as *P. xylostella*, *S. littoralis*, and *C. fumiferana* [151]. *B. thuringiensis* (Bt) strains have been developed as commercial biopesticides for more than a decade. Xia et al. found that the abundance of some bacteria in the larval midgut was related to the insecticide resistance of *P. xylostella.* Inoculating larvae with culturable gut microbes (*Enterobacter* sp. Mn2) reduced larval mortality after infection with *B. thuringiensis* in other studies, indicating that the gut microbiota can protect taxonomically diverse hosts from pathogen attack [51]. *Xenorhabdus nematophila* is another entomopathogenic bacterium that is symbiotically associated with parasitic nematodes (*Steinernema*). It is very effective against lepidopterans, such as the beet armyworm and diamondback moth [152,153].

In addition, *Wolbachia* species are widespread endosymbionts of lepidopteran insects. *Wolbachia* species, which are naturally occurring endosymbiotic bacteria found inside the cells of arthropods and filarial nematodes, can manipulate the host reproduction system [82] and are generally known as potential environmentally friendly biopesticides for the control of disease vectors and pests [154]. Recent studies found that *Wolbachia* species infect approximately 40% of terrestrial arthropod species, such as Lepidoptera, Hymenoptera, and Diptera species [155,156]. Cytoplasmic incompatibility (CI) is one of the most common phenotypes of reproductive manipulation in *Wolbachia* [157]. For instance, infection of *Homona magnanima* by multiple *Wolbachia* strains causes CI in the host, and *Wolbachia* increases the *H. magnanima* pupal weight and shortens the host development time [158]. Interestingly, *Wolbachia* spreads vertically in insects and is inherited maternally due to its presence in the cytoplasm of female gametes. Fukui et al.’s inhibition study of *Ostrinia* moths found that *Wolbachia* targets the population masculinization gene of the host to accomplish male killing by a failure of dosage compensation through unproductive mating [159].

These studies suggest that bacterial symbionts are essential in the evolution of insects, and thus, elucidating the role of bacterial symbionts of lepidopterans might help in the development of improved methods of biological control.

## 5. Conclusions: Implication and Outlook

The gut of lepidopteran insects is the primary site of digestion and absorption. It is the first line of defence against pathogens. With the development of molecular technologies such as high-throughput sequencing of the 16S rRNA gene and metagenome analysis, the research bottleneck has been overcome. Researchers can not only determine the classification and composition of the gut microbiota but also reveal the potential functions of symbionts in the host. The microbial community is closely related to host defence against pathogens. An artificial feeding system with gut wall cells cultured in vitro can be used to simulate the insect gut to evaluate the interactions of the host and gut microbiota.

In addition, we generalized the functions of the gut microbiota in lepidopteran insects (Table 1) to provide an overview of lepidopteran insect gut immune pathways (Figure 2). In this way, we can obtain a better understanding of the mechanisms of the gut microbiota. The established inventory of the microbiota of lepidopteran insects can be supplemented with a large amount of information on its taxonomy and genetics. These findings will help to clarify the microbiological composition of the lepidopteran insect gut and the specific functions of bacteria in such a specialized environment. On the other hand, if we alter the microecological composition of lepidopteran insects, for instance, by manipulating certain intestinal bacteria to improve their disease resistance ability and energy utilization through the digestive tract, enhanced biocontrol strategies against lepidopteran pests can be developed. In addition, the gut microbiota has the potential to be an environmentally friendly alternative control agent for diseases of economically important and beneficial insects and crops.

## Figures and Tables

**Figure 1 microorganisms-10-01234-f001:**
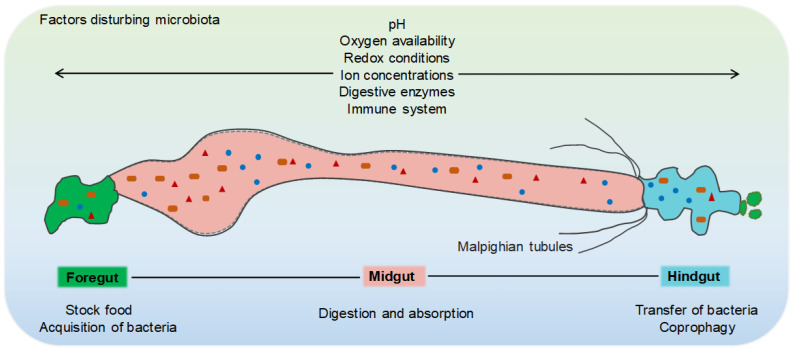
Gut structure of lepidopteran insects. Take the *B. mori* as an example. The foregut and hindgut are lined by a cuticle layer (thick black line), and the midgut secretes a peritrophic matrix (dashed line). Factors influencing the composition of the gut microbiota of lepidopteran insects include host development, pH, oxygen availability, redox conditions, ion concentrations, digestive enzymes and the immune system in different gut compartments, available sources for bacterial acquisition, and the capability to transfer bacteria to progeny. Green indicates the foregut, red indicates the midgut, and blue indicates the hindgut.

**Figure 2 microorganisms-10-01234-f002:**
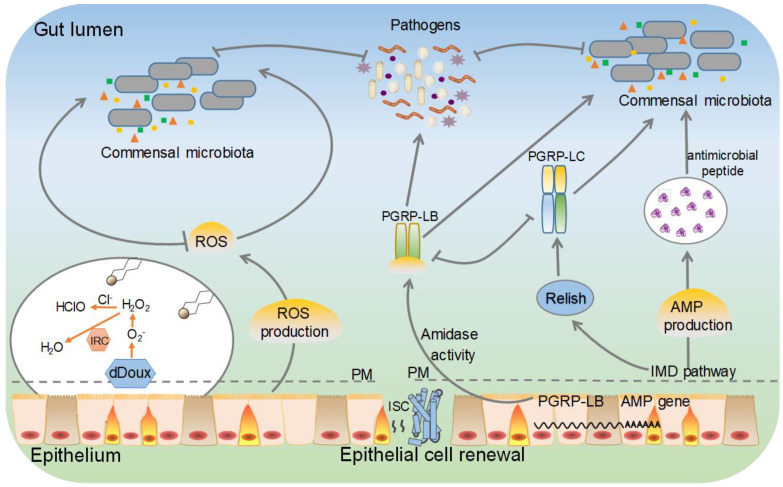
Putative immune signalling pathways are involved in the defences against pathogenic microbial infections in the gut of lepidopteran insects. This model is based on the local production of reactive oxygen species (ROS), and antimicrobial peptide (AMP) of *Drosophila* and findings in lepidopteran insects. The immune deficiency (IMD) includes the major signalling pathways inducing AMP production, and AMP genes provide inducible defense mechanisms in the gut. PM, peritrophic matrix.

**Table 1 microorganisms-10-01234-t001:** The category, function and reference for some important symbiotic bacteria and fungi of lepidopteran insects.

	Genus Level	Category	Function
**Bacteria**	*Bacillus*	Firmicutes	Counteract anti-herbivore plant defences [69]
	*Staphylococcus*	Firmicutes	Against plant-derived protease inhibitor; pest control [70]
	*Enterococcus*	Firmicutes	Increase anti-herbivore defence; insecticidal activities [71]
	*Methylobacterium*	Proteobacteria	Nitrogen fixation [72]
	*Sphingomonas*	Proteobacteria	Microbe-mediated detoxification of phytotoxins and pesticides [73]
	*Propionibacterium*	Actinobacteria	Produce antimicrobial peptides [74]
	*Microbacterium*	Actinobacteria	Antibiotic-resistant [75]
	*Pseudomonas*	Pseudomonas	Anti-phytopathogenic fungi [76]
	*Pantoea*	Proteobacteria	Affect oviposition behavior, morphogenesis and development [77,78]
	*Acinetobacter*	Proteobacteria	Metabolize insecticides [79]
	*Enterobacter*	Proteobacteria	Anti-phytopathogenic fungi activity; growth and development [80]
	*Wolbachia*	Proteobacteria	Participate in reproductive regulations, increase host resistance [81,82]
**Fungi**	*Stenotrophomonas*	Pseudomonadaceae	Insecticide resistance [83]
	*Cladosporium*	Ascomycota	Produce many antimicrobial agents [84]
	*Botrytis*	Ascomycota	Development and oviposition behavior [85]
	*Fusarium*	Ascomycota	Secretory defence [86]
	*Cryptococcus*	Basidiomycota	Immune Response [86]
	*Clonostachys*	Ascomycota	Potential biocontrol agents [87]
	*Erythrobasidium*	Basidiesvampar	Potential biocontrol agents [88]

## Data Availability

The data presented in this study are available in this article only.

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
