# Peer review of "Diversity and Functional Roles of the Gut Microbiota in Lepidopteran Insects"

_microorganisms, 2022, doi:10.3390/microorganisms10061234_

Round 1
Reviewer 1 Report
Review
of the manuscript “Diversity and Functional Roles of the Gut Microbiota in Lepidopteran
Insects”
by authors Xiancui Zhang, Fan Zhang and Xingmeng Lu
The authors present a detailed review of the literature data on the composition and functions of the gut microbiota of lepidopteran insects.
With the development of molecular technologies such as high-throughput sequencing of the 16S rRNA gene and metagenome analysis, researchers can not only determine the classification and composition of the gut microbiota but also reveal the potential functions of symbionts in the host.
Authors generalized the functions of the gut microbiota in Lepidopteran insects to provide an overview of lepidopteran insect gut immune pathways.
The generalization of the available data is also of practical importance, since the microbial community is closely related to host defence against pathogens.
Based on the created inventory of the microbiota of lepidopteran insects, biocontrol strategies against Lepidoptera pests can be developed. In addition, the gut microbiota may provide an environmentally friendly alternative means of controlling diseases of economically important and beneficial insects and crops.
The generalization carried out corresponds to the aims and objectives of the research. A detailed analysis of the literature sources was carried out. The list of references is quite complete. There is no excessive self-quoting. The submitted manuscript may be published as a Review in the Microorganisms journal.
Author Response
Thank you for carefully reading and commending our work.
Reviewer 2 Report
The manuscript is well written in general, with some issues which have to be clarified/added/changed prior to the final decision. Detailed info can be found in the attached pdf file.

Author Response
Response to Reviewer 2 Comments
Many thanks for your help to improve our manuscript. All these minor issues have been revised by “Track Changes” in the manuscript (MS Word), and comments have been included.
Point 1: Please give space before the citation.
Point 2: Please write down the full form as it appeared first here.
Point 3: italic
Point 4: this name has been mentioned several times above. Please write as P. xylostella.
Point 5: Please write down the abbreviated words in full in the caption.
Response: Thank you for your suggestion. We have checked the whole manuscript and revised similar types of problems.
Point 6: What is the 'similar composition'? Like Plutella xylostella? This is not clear.
Response 6: Thanks for your suggestion and we have modified it in lines 120-123.
Point 7: “few studies have addressed the gut microbiota composition of the reflected developmental process, especially in monophagous species.” Not clear. Please rewrite.
Response 7: According to your suggestions, we have redescribed this sentence. Detail is the “Furthermore, lepidopteran insects are holometabolic, and few studies have reflected the gut microbiota composition throughout development from egg to adult, especially in monophagous species.”
Point 8: “Next-generation sequencing of the B. mori strain Daizo revealed 13,769 expressed genes, and among them were known immunity genes with activity against BmCPV.” Not clear. Please check and rewrite.
Response 8: We are grateful for your suggestion and have redescribed this sentence. Detail is the “In addition, several immune-related genes were found to be implicated in midgut’s response against BmCPV infection of B. mori larval, including proteolytic enzymes, hormonal signaling, and heat-shock proteins.”